# Quantum Relational Knowledge Distillation

**Chen-Yu Liu**[1]    **Kuan-Cheng Chen**[2]    **Keisuke Murota**[1]
**Samuel Yen-Chi Chen**[3]    **Enrico Rinaldi**[1]
[1]Quantinuum, Partnership House, London, UK
[2]Imperial College London, London, UK
[3]Brookhaven National Laboratory, New York, US
{chen-yu.liu,keisuke.muroto,enrico.rinaldi}@quantinuum.com
ycchen1989@ieee.org
kuan-cheng.chen17@imperial.ac.uk

## Abstract

Knowledge distillation (KD) is a widely adopted technique for compressing large models into smaller, more efficient student models that can be deployed on devices with limited computational resources. Among various KD methods, Relational Knowledge Distillation (RKD) improves student performance by aligning relational structures in the feature space, such as pairwise distances and angles. In this work, we propose Quantum Relational Knowledge Distillation (QRKD), which extends RKD by incorporating quantum relational information. Specifically, we map classical features into a Hilbert space, interpret them as quantum states, and compute quantum kernel values to capture richer inter-sample relationships. These quantum-informed relations are then used to guide the distillation process. We evaluate QRKD on both vision and language tasks, including CNNs on MNIST and CIFAR-10, and GPT-2 on WikiText-2, Penn Treebank, and IMDB. Across all benchmarks, QRKD consistently improves student model performance compared to classical RKD. Importantly, both teacher and student models remain classical and deployable on standard hardware, with quantum computation required only during training. This work presents the first demonstration of quantum-enhanced knowledge distillation in a fully classical deployment setting.

## 1 Introduction

Modern machine learning has been driven by the success of large-scale models, such as deep convolutional networks and large language models (LLMs) [1, 2]. These models often achieve state-of-the-art performance across domains ranging from computer vision to natural language processing. However, their exceptional accuracy typically comes at the cost of significant computational and memory demands. Deploying such large models on resource-constrained edge devices or latency-sensitive applications remains a major challenge [3].

To address this issue, knowledge distillation (KD) has emerged as a powerful model compression technique [4]. In KD, a compact student model is trained to mimic the behavior of a larger teacher model, typically by matching the output logits or soft predictions. This enables the student to inherit the teacher's knowledge while retaining a smaller and more efficient architecture. Despite its empirical success, conventional KD often overlooks internal representational structures learned by the teacher [5–8].

Relational Knowledge Distillation (RKD) [8] improves upon this by encouraging the student to match not just the teacher's outputs but also the relational structure of its internal representations. By aligning pairwise distances and/or angles between sample embeddings in the feature space, RKD

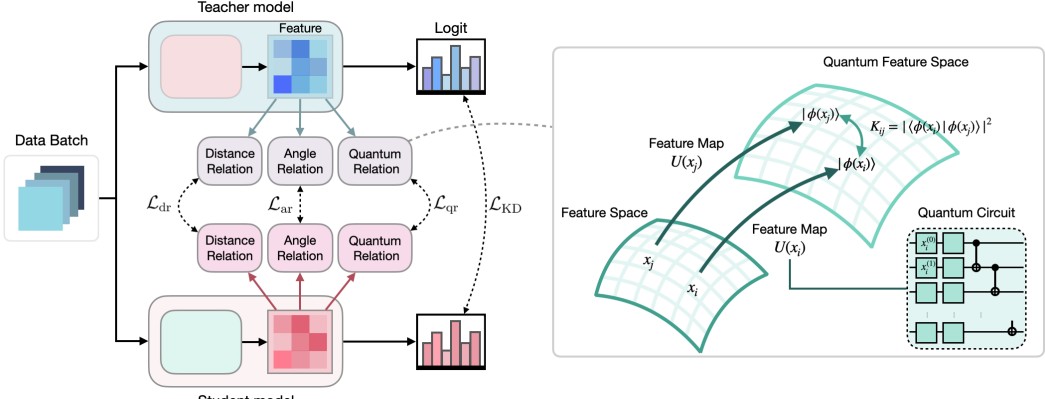

Figure 1: **Overview of the proposed QRKD framework.** A batch of data is fed through both the teacher and student models to extract intermediate features. From these, we compute pairwise distance relations and angle relations to construct the relational losses $\mathcal{L}_{dr}$ and $\mathcal{L}_{ar}$, following the RKD framework. Additionally, both teacher and student features are embedded into a Hilbert space (quantum feature space) via parameterized quantum circuits $U(x)$, yielding quantum states $|\phi(x)\rangle$. The pairwise quantum relations are evaluated via fidelity kernels $K_{ij} = |\langle\phi(x_i)|\phi(x_j)\rangle|^2$ (in this example), forming the quantum alignment loss $\mathcal{L}_{qr}$. Finally, logit outputs are used to compute the knowledge distillation loss $\mathcal{L}_{KD}$. All losses are combined to guide student training under the QRKD objective. The right panel illustrates how classical features are mapped into the quantum feature space via encoding circuits and evaluated by quantum kernel functions.

guides the student to mimic the geometric organization of the teacher's learned space, offering stronger generalization and better functional imitation.

Yet, this approach is inherently restricted to the capacity of the classical feature space. A natural question arises: can we distill knowledge more effectively by aligning teacher and student in a richer, higher-dimensional space? If the feature space is too limited to capture complex relations, the distillation process itself may become bottlenecked. Expanding beyond the classical setting could provide a new path for enhancing model compression.

Quantum computing, and in particular quantum machine learning (QML) [9, 10], offers a promising direction for enhancing model compression through high-dimensional feature representations [11, 12]. Quantum systems naturally operate in Hilbert spaces whose dimensionality grows exponentially with the number of qubits ($\mathbb{C}^{2^n}$ for $n$ qubits), providing access to expressive function classes that are difficult to simulate classically. Quantum kernel methods leverage this by embedding classical data into quantum states via parameterized quantum circuits, enabling rich nonlinear transformations. The resulting quantum kernels, computed either through inner products between quantum states (fidelity kernels) [13, 14] or via projected measurements on subsystems (projected quantum kernels) [15], provide flexible and expressive similarity metrics in high-dimensional Hilbert spaces, making them well-suited for learning geometry and relational alignment.

In this context, the quantum kernel value can be interpreted as a relational measure between data points. Rather than aligning features in the original space, we propose to align the relational structure of teacher and student models in the quantum feature space. This leads to a natural extension of RKD into the quantum domain.

We propose Quantum Relational Knowledge Distillation (QRKD), a new method that maps intermediate features of both teacher and student models into quantum states via parameterized quantum circuits, and aligns their relational structures by minimizing the discrepancy between their pairwise quantum kernel values. By leveraging the expressivity of high-dimensional Hilbert space embeddings, QRKD enables knowledge transfer in a geometrically rich space that may be computationally expensive to simulate classically when scaled up[1].

---

[1]Many existing quantum kernels have been shown to admit efficient classical approximations, but a general dequantization proof for all quantum kernels is still lacking [16, 17].

In classical approaches, mapping a feature vector $h(x) \in \mathbb{R}^d$ into a higher-dimensional space $\mathbb{R}^D$ typically requires $O(d \cdot D)$ parameters for a linear projection, where $D$ is the target dimensionality and often much larger than $d$. In contrast, QRKD uses a parameterized quantum circuit to embed features into a $2^n$-dimensional Hilbert space using only $O(d)$ parameters in the quantum circuit, and $n$ is the number of qubits with $2^n \sim D$. This parameter efficiency allows QRKD to access rich relational structures in high-dimensional quantum feature spaces with significantly reduced overhead compared to classical alternatives.

Our main contributions are:

1. **Quantum Relational Knowledge Distillation (QRKD)**: We introduce QRKD, a hybrid quantum-classical distillation framework that aligns intermediate representations of teacher and student models in the Hilbert space through quantum kernel alignment. This enables geometric regularization of the distillation process in an exponentially large quantum feature space.

2. **Theoretical Insights**: We provide theoretical insights showing that, under mild assumptions such as injectivity and Lipschitz continuity, aligning pairwise quantum kernel values between teacher and student leads to functional closeness. This generalizes the classical RKD formulation to the quantum setting and gives a principled justification for enhanced distillation.

3. **Empirical Validation and Compatibility**: We demonstrate empirically that QRKD consistently improves student model performance and narrows the performance gap between teacher and student more effectively than conventional KD or RKD methods. QRKD is fully compatible with standard distillation losses (e.g., KL divergence on soft logits) and can be seamlessly integrated into existing distillation pipelines with minimal architectural changes.

4. **Classical Inference Compatibility**: Although QRKD leverages quantum computation during training to extract relational knowledge in Hilbert space, the resulting student model remains purely classical and can be deployed on conventional hardware without requiring quantum resources at inference time.

These results highlight the potential of quantum feature space alignment as a new principle for knowledge distillation, with promising applications in quantum-enhanced model compression and hybrid learning systems.

## 2 Related Works

### 2.1 Knowledge Distillation

The concept of KD was first introduced in [4], where the teacher-student framework was proposed. In this setting, a large, pre-trained teacher model transfers knowledge to a smaller student model by providing soft targets, defined as the class probability outputs of the teacher, rather than hard labels. These soft targets contain richer supervisory information by capturing inter-class relationships. For example, the KD objective to be minimized can be formulated as

$$\mathcal{L}_{\text{KD}} = \text{KL}\left(\text{softmax}\left(\frac{f_t(x_i)}{\tau}\right), \text{softmax}\left(\frac{f_s(x_i)}{\tau}\right)\right), \tag{1}$$

where $\text{KL}(\cdot, \cdot)$ denotes the Kullback–Leibler divergence, $\tau$ is the temperature parameter controlling the smoothness of the output distributions, and $f_t$, $f_s$ are the pre-softmax (logit) outputs of the teacher and student models, respectively. This process is commonly referred to as logit matching. While many variants of KD have been proposed [5–8], we focus specifically on Relational Knowledge Distillation (RKD) [8], as its underlying principles are particularly well-suited for potential extensions into the quantum domain.

### 2.2 Relational Knowledge Distillation

Building on the standard KD framework, RKD focuses on transferring the structural relationships among examples in the feature space, rather than solely aligning the teacher and student outputs. Let $h_t(x), h_s(x) \in \mathbb{R}^d$ denote the intermediate feature representations of the teacher and student models,

respectively, and let $f(x) = g(h(x))$ represent the final output, where $g(\cdot)$ is a downstream prediction head. RKD encourages the student to preserve structural relations between feature representations. For example, the distance-based RKD loss is defined as:

$$\mathcal{L}_{\text{dr}} = \sum_{i,j} \ell_\delta \left( d_s(x_i, x_j) - d_t(x_i, x_j) \right), \tag{2}$$

where $d(x_i, x_j) = \|h(x_i) - h(x_j)\|$ and $\ell_\delta$ denotes the Huber loss. Assume the output function $g(\cdot)$ is Lipschitz with respect to the intermediate features, i.e.,

$$\|f_s(x) - f_t(x)\| \leq L \cdot \|h_s(x) - h_t(x)\|. \tag{3}$$

Now suppose the RKD loss satisfies the following distance alignment bound:

$$\sum_{i,j} \left( \|h_s(x_i) - h_s(x_j)\| - \|h_t(x_i) - h_t(x_j)\| \right)^2 \leq \epsilon. \tag{4}$$

Under mild assumptions, such as feature normalization and bounded norms, this implies that:

$$\frac{1}{n} \sum_{i=1}^{n} \|h_s(x_i) - h_t(x_i)\|^2 \leq C \cdot \epsilon, \tag{5}$$

for some constant $C > 0$ depending on the feature geometry. By Lipschitz continuity, this further implies a bound on output discrepancy:

$$\mathbb{E}_{x \sim \mathcal{D}} \left[ \|f_s(x) - f_t(x)\|^2 \right] \leq L^2 \cdot \mathbb{E}_{x \sim \mathcal{D}} \left[ \|h_s(x) - h_t(x)\|^2 \right] \leq O(\mathcal{L}_{\text{dr}}). \tag{6}$$

This reasoning extends naturally to other relational objectives, such as angle-based RKD losses $\mathcal{L}_{\text{ar}}$ [8]. Overall, the analysis shows that minimizing relational losses, such as pairwise distances or angles, promotes output-level alignment between the student and teacher, even in the absence of direct supervision on the output layer. As a result, RKD can be viewed as a form of functional imitation guided by structural constraints.

## 2.3  Quantum Kernel Methods

Quantum kernel methods employ quantum circuits to construct feature maps into high-dimensional Hilbert spaces, enabling nonlinear learning through quantum-enhanced similarity measures. This follows recent work on kernel-based representation similarity and relational knowledge transfer [8, 18–20], but extends it into the quantum domain by encouraging consistent quantum feature geometry. A quantum kernel is typically defined as

$$k(x_i, x_j) = F \left( |\phi(x_i)\rangle, |\phi(x_j)\rangle \right), \tag{7}$$

where $|\phi(\mathbf{x})\rangle = U(\mathbf{x})|0\rangle^{\otimes n}$ represents a quantum state generated by a parameterized quantum circuit $U(\mathbf{x})$, commonly referred to as an encoding circuit based on the classical input $x$. The function $F$ quantifies the similarity between two quantum states, serving as the kernel function. To construct the full quantum state used in our method, we apply $R_y$ rotation per qubit across multiple layers, followed by a chain of CNOT gates to introduce entanglement. The resulting encoded quantum state is expressed as:

$$|\phi(\mathbf{x})\rangle = U(\mathbf{x})|0\rangle^{\otimes n} = \left[ \left( \prod_{k=1}^{n-1} \text{CNOT}^{k,k+1} \right) \left( \prod_{l=1}^{L} \prod_{j=1}^{n} R_y^j(x^{j+nl}) \right) \right]^{L_N} |0\rangle^{\otimes n}, \tag{8}$$

where $n$ is the number of qubits, $L$ is the number of encoding layers, $L_N$ is the number of repetitions, and $R_y^j$ denotes the $R_y$ rotation applied to the $j$-th qubit, $x^{j+nl}$ denotes the $j + nl$-th element in $\mathbf{x}$. A schematic diagram of this encoding circuit is provided in Fig. 2. Thus, if the input vector $\mathbf{x}$ has dimension $d$, the corresponding quantum circuit maps it into a quantum state residing in a Hilbert space of dimension $2^n$, where $n$ is the number of qubits.

Two main classes of quantum kernels have been widely studied: fidelity-based kernels [13], which directly measure state overlap, and projected kernels [15], which derive similarity from local measurement statistics.

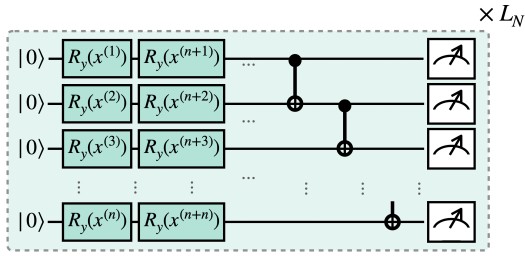

Figure 2: **Illustration of the angle encoding circuit we use in the QRKD.**

**Fidelity Quantum Kernel.** The fidelity quantum kernel (FQK), also known as the embedding kernel or quantum state overlap, is defined as

$$k_{\text{fid}}(x_i, x_j) = |\langle\phi(x_i)|\phi(x_j)\rangle|^2,$$

and measures the squared inner product between quantum states, defined by a parameterized quantum circuit. Quantum kernel methods typically assume that $|\phi(x)\rangle$ is implemented by a circuit of polynomial depth in the input size, enabling efficient state preparation and kernel evaluation on quantum hardware [14, 13, 21]. The resulting kernel value $k(x_i, x_j) = |\langle\phi(x_i)|\phi(x_j)\rangle|^2$ can be estimated using subroutines such as the SWAP test or interference-based circuits. Since $|\phi(x)\rangle$ embeds data into a Hilbert space of exponential dimension $2^n$ with $n$ qubits, fidelity kernels provide access to rich, high-capacity feature spaces using only polynomial computational resources.

However, fidelity kernels suffer from a significant limitation: as both circuit width and depth increase, the pairwise kernel values tend to concentrate around a constant due to high-dimensional geometry. This phenomenon, known as *exponential concentration*, limits the expressivity and generalization capacity of deep quantum kernel models [22]. Thus, while fidelity kernels are expressive in principle, their performance can degrade in practical, high-depth or noisy regimes.

**Projected Quantum Kernel.** In contrast to fidelity kernels that rely on global measurements over the entire quantum state, recent works have introduced the concept of *projected quantum kernels (PQK)* [15], which define similarity measures based on partial information obtained from local observables. Instead of computing the fidelity between full quantum states, PQK evaluates measurements on selected subsystems, such as Pauli-Z observables on individual qubits, and constructs classical feature vectors from these outcomes. A representative formulation, proposed in [23], defines the projected kernel as

$$k_{\text{proj}}(x_i, x_j) = \text{tr}\left[\rho_{(1)}(x_i)\rho_{(1)}(x_j)\right], \tag{9}$$

where $\rho_{(1)}(x) = \text{tr}_{j \neq 1}(|\phi(x)\rangle\langle\phi(x)|)$ is the one-qubit reduced density matrix obtained by tracing out all but the first qubit from the encoded quantum state $|\phi(x)\rangle$.

PQKs have been shown to retain expressivity while avoiding the global-measurement-induced concentration of fidelity kernels, which causes most pairwise similarities to collapse toward a constant under deep or unstructured circuits [22]. Furthermore, data-dependent pretraining techniques have been shown to mitigate or even eliminate the concentration effect in the PQK setting [23]. These properties make PQK especially suitable for hybrid quantum-classical learning and for geometric regularization strategies such as those employed in our proposed QRKD method.

## 3 Relational Knowledge Distillation in Quantum Feature Space

Building on the RKD framework introduced in the previous section, we extend relational distillation into the quantum domain by leveraging quantum feature embeddings and pairwise kernel alignment, to propose **Quantum Relational Knowledge Distillation (QRKD)**. As before, let $h_t(x), h_s(x) \in \mathbb{R}^d$ denote the intermediate representations of the teacher and student with input data $x$. These are encoded into quantum states via a parameterized quantum circuit (Eq. 8):

$$|\phi(h(x))\rangle = U(h(x))|0\rangle^{\otimes n},$$

where $U(h(x))$ is a data-dependent unitary and $|\phi(h(x))\rangle \in \mathcal{H}_q$ lies in a high-dimensional quantum Hilbert space with qubit number $n$. To capture the relational structure between embedded features for

different input data, we compute the fidelity kernel for example:

$$k(x_i, x_j) = |\langle \phi(h(x_i)) \mid \phi(h(x_j)) \rangle|^2,$$

which measures the squared inner product (i.e., fidelity) between two quantum states. The QRKD loss is then defined by aligning teacher and student kernel values across sample pairs:

$$\mathcal{L}_{\mathrm{qr}} = \sum_{i,j} \ell_\delta \left( k_s(x_i, x_j) - k_t(x_i, x_j) \right), \tag{10}$$

where $k_t, k_s$ denote the fidelity kernels derived from the teacher and student embeddings, respectively. Assuming the quantum feature map $\phi$ is injective and approximately distance-preserving[2], kernel alignment implies geometric closeness between the original feature vectors. Combined with the same Lipschitz assumption on the output function $f(x) = g(h(x))$, we obtain the following bounds:

$$\sum_{i,j} \left( k_s(x_i, x_j) - k_t(x_i, x_j) \right)^2 \leq \epsilon \tag{11}$$

$$\Rightarrow \quad \frac{1}{n} \sum_{i=1}^{n} \| h_s(x_i) - h_t(x_i) \|^2 \leq C' \cdot \epsilon \tag{12}$$

$$\Rightarrow \quad \mathbb{E}_{x \sim \mathcal{D}} \left[ \| f_s(x) - f_t(x) \|^2 \right] \leq L^2 \cdot C' \cdot \epsilon = O(\mathcal{L}_{\mathrm{qr}}), \tag{13}$$

where $C' > 0$ depends on the embedding geometry and the expressivity of $\phi$. In this case, QRKD encourages functional alignment between teacher and student by enforcing similarity of quantum-encoded relational structures. Unlike conventional RKD, QRKD operates in a quantum-induced feature space, offering higher representational flexibility while preserving a clear geometric interpretation.

After introducing all components of the QRKD framework, we define the overall training objective for the student model as a weighted combination of task supervision, knowledge distillation, relational losses, and quantum kernel alignment:

$$\mathcal{L}_{\mathrm{QRKD}} = \alpha \cdot \mathcal{L}_{\mathrm{task}} + \beta \cdot \mathcal{L}_{\mathrm{KD}} + \gamma_{\mathrm{d,a}} \cdot (\mathcal{L}_{\mathrm{dr}} + \mathcal{L}_{\mathrm{ar}}) + \omega \cdot \mathcal{L}_{\mathrm{qr}}, \tag{14}$$

where $\mathcal{L}_{\mathrm{task}}$ denotes the task-specific loss (e.g., cross-entropy for classification or text generation in our examples), and $\mathcal{L}_{\mathrm{KD}}$ corresponds to the logit-based distillation loss defined in Eq. 1. The terms $\mathcal{L}_{\mathrm{dr}}$ and $\mathcal{L}_{\mathrm{ar}}$ represent the relational distance and angle losses, respectively, as introduced in Section 2.2. Finally, $\mathcal{L}_{\mathrm{qr}}$ is the quantum kernel alignment loss described in Eq. 10. The coefficients $\alpha$, $\beta$, $\gamma_{\mathrm{d,a}}$, $\omega \in \mathbb{R} \geq 0$ are tunable hyperparameters that control the contribution of each component.

## 4 Empirical Experiments

### 4.1 Experiment Setup

To evaluate the effectiveness of the proposed QRKD method, we conduct experiments across both vision and language domains. Our goal is to demonstrate that QRKD provides consistent benefits across a variety of architectures and tasks, beyond the original image classification setting used in RKD [8]. We evaluate QRKD on the MNIST and CIFAR-10 datasets using convolutional neural networks (CNNs). For MNIST, we adopt a VGG-style CNN with 6,690 parameters as the teacher and a smaller variant with 1,725 parameters as the student. For CIFAR-10, we use a similar CNN structure scaled to 277,610 parameters for the teacher and 79,946 parameters for the student, to account for the dataset's increased complexity.

To assess QRKD in natural language processing (NLP) and LLM settings, we apply it to the WikiText-2 [26], Penn Treebank [27], and IMDB [28] datasets using GPT-2 [2] as the backbone. The teacher

---

[2]The term "approximately distance-preserving" acknowledges that the quantum feature map may intentionally distort geometric relationships when embedding classical features into a high-dimensional Hilbert space. This distortion is a feature shared with classical kernel methods: with high probability [24], such mappings can transform nonlinearly separable structures in the input space into linearly separable ones in the feature space [25, 13]. While exact distance preservation is not required, the quantum feature map often retains sufficient local or task-relevant geometry such that kernel alignment in Hilbert space still implies meaningful similarity between the original representations.

model is a standard GPT-2 with 124M parameters, configured with `n_layer = 12`, `n_head = 12`, and `n_embd = 768`. The student model has approximately 30M parameters, using `n_layer = 6`, `n_head = 6`, and `n_embd = 384`. The quantum circuit uses angle encoding with $R_y$ rotations followed by a chain of CNOT gates to introduce entanglement across qubits. Each input feature vector is partitioned evenly across qubits and encoded into the circuit through layered parameterized rotations (Eq. 8). In all experiments in the main text, we set the number of qubits to $n = 4$, the number of repetitions $L_n = 2$ for fidelity kernel, and $L_n = 1$ for projected kernel.

## 4.2 General performance

**Image Classification Tasks.** We evaluate the performance of QRKD method and its variants against standard baselines on the MNIST dataset. Table. 1 reports training and testing accuracy, accuracy generalization gap (Acc. Gap), teacher-student (T&S) gap, and distillation gain. The accuracy generalization gap measures the difference between training and test accuracy. The T&S gap quantifies the test accuracy difference between the teacher and student models, while the distillation gain quantifies the improvement in student performance due to distillation compared to training from scratch. The student model trained from scratch (F. Scratch) achieves a strong baseline with a test accuracy of 94.91%. Traditional logit-based KD underperforms significantly, yielding a lower test accuracy (89.24%) and a large T&S gap (9.46), indicating poor generalization and ineffective knowledge transfer. RKD improves upon KD by leveraging structural relationships between samples, achieving 95.07% test accuracy and a reduced T&S gap of 3.63. Our proposed QRKD method further enhances performance, attaining the highest test accuracy of 95.46%, the lowest T&S gap (3.24), and the largest distillation gain (0.55), demonstrating the effectiveness of incorporating quantum-inspired relational cues. To better understand the contribution of each component in QRKD, we evaluate three ablations: QRKD-A (quantum + angle loss), QRKD-D (quantum + distance loss), and QRKD-Q (quantum loss only).

Table 1: Performance comparison of KD, RKD, and QRKD on MNIST. Note: Bold values indicate the best performance, while † marks the second-best result in each column. 5 samples are used to calculate the mean and variance.

| MNIST ($L_N = 2$) | | | | | |
|---|---|---|---|---|---|
| | Train ↑ | Test ↑ | Acc. Gap ↓ | T&S Gap ↓ | Dist. Gain ↑ |
| F. scratch | $94.79 \pm 2.02$ | $94.91 \pm 2.03$ | -0.11 | 3.79 | - |
| KD | $89.04 \pm 16.53$ | $89.24 \pm 16.47$ | **-0.20** | 9.46 | -5.67 |
| RKD | $\mathbf{94.88 \pm 2.32}^{\dagger}$ | $\mathbf{95.07 \pm 2.31}^{\dagger}$ | $-0.19^{\dagger}$ | $\mathbf{3.63}^{\dagger}$ | $\mathbf{0.16}^{\dagger}$ |
| QRKD | $\mathbf{95.29 \pm 1.34}$ | $\mathbf{95.46 \pm 1.32}$ | -0.17 | **3.24** | **0.55** |
| QRKD-A | $93.83 \pm 4.01$ | $94.02 \pm 4.23$ | -0.18 | 4.68 | -0.89 |
| QRKD-D | $90.22 \pm 16.87$ | $90.34 \pm 16.94$ | -0.12 | 8.36 | -4.56 |
| QRKD-Q | $89.62 \pm 16.82$ | $89.79 \pm 17.05$ | -0.17 | 8.90 | -5.11 |
| Teacher | 99.32 | 98.70 | 0.62 | - | - |

Table 2: Performance comparison of KD, RKD, and QRKD on CIFAR-10.

| CIFAR-10 ($L_N = 2$) | | | | | |
|---|---|---|---|---|---|
| | Train ↑ | Test ↑ | Acc. Gap ↓ | T&S Gap ↓ | Dist. Gain ↑ |
| F. scratch | $\mathbf{75.89 \pm 0.76}$ | $65.59 \pm 0.32$ | 10.29 | 1.09 | - |
| KD | $74.16 \pm 0.94$ | $65.66 \pm 0.72$ | **8.50** | 1.02 | 0.07 |
| RKD | $74.50 \pm 0.65$ | $65.47 \pm 0.54$ | 9.03 | 1.21 | -0.11 |
| QRKD | $\mathbf{74.94 \pm 0.60}^{\dagger}$ | $65.58 \pm 0.43$ | 9.36 | 1.10 | 0.01 |
| QRKD-A | $74.62 \pm 0.40$ | $\mathbf{65.72 \pm 0.64}$ | 8.90 | **0.96** | **0.13** |
| QRKD-D | $74.22 \pm 0.82$ | $\mathbf{65.67 \pm 0.81}^{\dagger}$ | 8.55 | $\mathbf{1.01}^{\dagger}$ | $\mathbf{0.08}^{\dagger}$ |
| QRKD-Q | $74.16 \pm 0.94$ | $65.66 \pm 0.72$ | **8.50** | 1.02 | 0.07 |
| Teacher | 91.13 | 66.68 | 24.45 | - | - |

Following the MNIST evaluation, we assess the same set of distillation strategies on the more challenging CIFAR-10 dataset. The results are summarized in Table 2. The student trained from scratch reaches a test accuracy of 65.59%, which serves as the baseline. Among the classical baselines, KD achieves a slightly higher test accuracy (65.66%) with the lowest accuracy gap (8.50) and the

third-best teacher–student (T&S) gap (1.02), yielding a positive distillation gain of 0.07. This suggests that logit-based supervision remains effective in this setting.

While RKD was competitive on MNIST, it performs slightly worse here, with a lower test accuracy (65.47%) and a negative distillation gain (–0.11), indicating that relational constraints based on pairwise distances and angles may be less effective in complex visual domains.

Notably, the proposed QRKD variants demonstrate consistent improvements over standard KD and RKD baselines. Among them, QRKD-A achieves the best test accuracy (65.72%) and the lowest T&S gap (0.96), while also attaining the highest distillation gain (0.13), indicating more stable teacher–student alignment. QRKD-D similarly performs competitively (65.67% test accuracy) with distillation efficiency (0.08). Overall, QRKD and its variants outperform RKD and match or surpass classical KD, showing that incorporating quantum-inspired relational alignment yields tangible benefits even on a complex dataset like CIFAR-10, while different QRKD designs highlight distinct trade-offs between generalization and relational consistency.

**Text Generation Tasks.** We used perplexity as a metric to evaluate how well a probabilistic model predicts a given text dataset. Table 3 reports the testing perplexity across three benchmark datasets: WikiText-2, Penn Treebank (PTB), and IMDB. The evaluated methods include standard fine-tuning (FT), KD, RKD, our proposed QRKD, and its ablations (QRKD-A, QRKD-D, QRKD-Q).

Unlike the image classification experiments, where we primarily used the FQK and reported the PQK results for MNIST in Appendix D, the text generation experiments incorporate both FQK and PQK. Detailed results for PQK are presented in the following paragraphs, and PQK is used consistently across all QRKD variants in the text generation setting.

Table 3: GPT-2 testing perplexity on the text generation task using FT, KD, RKD, and QRKD variants across the WikiText-2, Penn Treebank, and IMDB datasets, using the quantum fidelity kernel ($L_N = 2$).

| | Teacher | FT | KD | RKD | QRKD | QRKD-A | QRKD-D | QRKD-Q |
|---|---|---|---|---|---|---|---|---|
| WikiText-2 ↓ | 1.5362 | 2.0029 | 1.9573 | 2.0183 | 2.3052 | **1.9520**[†] | 2.3331 | **1.9515** |
| Penn Treebank ↓ | 1.1698 | 1.2318 | 1.2203 | 1.2303 | 1.2638 | **1.2197** | 1.2876 | **1.2201**[†] |
| IMDB ↓ | 5.8707 | 12.3488 | **11.0990**[†] | 12.1082 | 15.5784 | 11.2012 | 15.3444 | **11.0886** |
| Avg ↓ | 2.8589 | 5.1945 | **4.7588** | 5.1189 | 6.3824 | 4.7909 | 6.3217 | **4.7534** |
| Norm. Avg ↓ | 1 | 1.4867 | **1.4026**[†] | 1.4760 | 1.7448 | 1.4071 | 1.7443 | **1.4007** |
| Norm. T&S Gap ↓ | - | 0.4867 | **0.4026**[†] | 0.4760 | 0.7448 | 0.4071 | 0.7443 | **0.4007** |
| Norm. Dist. Gain ↑ | - | - | **0.0841**[†] | 0.0107 | -0.2581 | 0.0796 | -0.2575 | **0.0866** |

*Fidelity Quantum Kernel.* – As expected, the teacher model consistently achieves the lowest perplexity across all datasets and serves as the upper bound. Among student models, QRKD-Q performs competitively on WikiText-2, achieving the second-best perplexity 1.9547, closely following the teacher. On PTB, both QRKD-A and QRKD-Q yield near identical best perplexity scores (1.2197), indicating that angle and quantum components contribute similarly in this setting. On the IMDB dataset, QRKD-Q achieves the best student performance with a perplexity of 11.0925, outperforming both RKD and KD.

Table 4: GPT-2 testing perplexity on the text generation task using FT, KD, RKD, and QRKD variants across the WikiText-2, Penn Treebank, and IMDB datasets, using the quantum projected kernel ($L_N = 1$).

| | Teacher | FT | KD | RKD | QRKD | QRKD-A | QRKD-D | QRKD-Q |
|---|---|---|---|---|---|---|---|---|
| WikiText-2 ↓ | 1.5362 | 2.0029 | **1.9573**[†] | 2.0183 | 2.3912 | **1.9520** | 2.3331 | 1.9621 |
| Penn Treebank ↓ | 1.1698 | 1.2318 | **1.2203**[†] | 1.2303 | 1.2838 | **1.2197** | 1.2876 | 1.2204 |
| IMDB ↓ | 5.8707 | 12.3488 | **11.0990** | 12.1082 | 14.8875 | 11.2012 | 15.3444 | **11.1332**[†] |
| Avg ↓ | 2.8589 | 5.1945 | **4.7588** | 5.1189 | 6.1875 | 4.7909 | 6.3217 | **4.7719**[†] |
| Norm. Avg ↓ | 1 | 1.4867 | **1.4026** | 1.4760 | 1.7299 | 1.4071 | 1.7443 | **1.4056**[†] |
| Norm. T&S Gap ↓ | - | 0.4867 | **0.4026** | 0.4760 | 0.7299 | 0.4071 | 0.7443 | **0.4056**[†] |
| Norm. Dist. Gain ↑ | - | - | **0.0841** | 0.0107 | -0.2431 | 0.0796 | -0.2575 | **0.0811**[†] |

When considering overall performance across all datasets, QRKD-Q achieves the best normalized average perplexity (1.4015), outperforming all other student models. This highlights the strength of

the quantum loss component in generalizing across diverse text generation tasks. KD follows closely with a normalized average perplexity of 1.4026. These findings suggest that QRKD-Q, despite its simplicity, is a strong candidate for general-purpose distillation in generative language modeling.

*Projected Quantum Kernel.* – While QRKD-Q demonstrates strong performance across datasets when using the FQK, its performance is less dominant when PQK is used as the quantum relational measure, as shown in Table 4. In this setting, QRKD-Q achieves a normalized average perplexity of 1.4056, a normalized teacher–student (T&S) gap of 0.4056, and a positive distillation gain of 0.0811. These results indicate that QRKD-Q remains competitive and maintains strong generalization across datasets; however, traditional KD still exhibits a slight advantage in overall performance efficiency for text generation tasks under the PQK setup.

Although the results in this section are evaluated using a fixed number of qubits for each task, we further explore the impact of qubit count in Appendix D. In this analysis, we examine how varying the number of qubits, which corresponds to different allocations of quantum resources, influences both model performance and the variance of the kernel values. This involves encoding the input features into quantum circuits of varying sizes. For the MNIST classification task, we observe a clear trend: increasing the number of qubits consistently improves performance. In contrast, for the text generation task using GPT-2 on the WikiText-2 dataset, the relationship is more nuanced. Each QRKD variant exhibits an optimal qubit count, beyond which performance may plateau or decline. This suggests that the effectiveness of quantum resource allocation is task-dependent and must be carefully tuned for different domains.

## 5 Discussion and Conclusion

In this work, we proposed **Quantum Relational Knowledge Distillation (QRKD)**, a novel hybrid framework that introduces quantum kernel-based relational alignment into the knowledge distillation process. By embedding intermediate features from both the teacher and student into a quantum Hilbert space and aligning their quantum kernel values, QRKD enhances the student's ability to approximate the teacher's behavior while preserving structural similarity in a quantum feature space.

While QRKD offers a theoretically motivated and empirically validated approach to improve distillation, several practical limitations remain. Most notably, the method requires quantum computations during training to evaluate quantum kernel values. Although we use idealized simulations in this work as a proof-of-concept, future deployment will face challenges from hardware noise and sampling variance, especially for fidelity-based kernels (see Appendix D). Extending QRKD to noisy quantum processors and designing robust circuits or measurement schemes are promising directions for future work. Furthermore, a comparison between QRKD and classical kernel methods is provided in Appendix F.

Importantly, QRKD maintains a fully classical inference pipeline. Both the teacher and student models are classical neural network models, and quantum resources are used only during training for relational supervision. This ensures compatibility with existing classical deployment infrastructure and avoids the need for quantum hardware during inference.

Compared to prior studies on quantum KD language models [29] (More in Appendix B), which often focus on binary classification or synthetic benchmarks, QRKD goes further by addressing the task of language modeling using real-world datasets and transformer-based architectures. This shift represents a meaningful step toward integrating quantum KD methods into high-level language modeling.

Our findings suggest that quantum relational alignment can serve as an effective and general-purpose tool for compressing large models and transferring knowledge, particularly in settings where parameter efficiency and training generalization are critical. We hope this work opens new avenues for exploring quantum-enhanced knowledge distillation, model compression, and hybrid quantum-classical learning for both vision and language domains.

## Acknowledgements

We are grateful to Stephen Clark and Harry Buhrman for insightful and invaluable discussions.

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

# Appendix

## A Quantum Machine Learning.

While classical machine learning (ML) has achieved remarkable success across a wide range of domains [30–36], quantum machine learning (QML) is also advancing rapidly. QML investigates how quantum computing can enhance machine learning by exploiting principles such as superposition and entanglement to potentially improve training efficiency and model performance. For example, quantum neural networks (QNNs) have been proposed as a means to accelerate learning through quantum parallelism and increased model expressivity [10, 9, 37, 38]. In addition to QNNs, quantum kernel methods have emerged as a prominent approach. These methods construct kernels based on inner products between quantum states, allowing classical data to be embedded into high-dimensional Hilbert spaces using quantum circuits [39]. The resulting quantum kernels can provide expressive similarity measures that are difficult to approximate using classical resources. Furthermore, the incorporation of Grover's search algorithm into QML pipelines has shown potential improvements in certain classification tasks [40, 41]. QML is considered a promising tool for tackling complex and high-dimensional data, with growing applications in areas such as drug discovery, stellar classification, natural language processing, recommender systems, and quantum generative modeling [42–50].

While promising in theory, QML is still far from mature due to a number of unresolved limitations that hinder its practical deployment. Among the foremost issues are the challenges related to model learnability and trainability [51–59], which raise concerns about the effectiveness and reliability of current QML architectures. From a practical perspective, many QML implementations rely on gate-based angle encoding to map classical data into quantum states. This strategy, however, struggles with scalability. As the size of the input data grows, the associated quantum circuits must also increase in both width and depth, which exacerbates error rates and limits performance on current noisy intermediate-scale quantum (NISQ) hardware [60]. To address these limitations, dimensionality reduction is often applied as a preprocessing step. Yet, this comes at the cost of potentially discarding critical information. For QML to match the versatility of classical neural networks, it must be capable of processing high-dimensional inputs without relying solely on aggressive compression. This highlights the need for more effective quantum-compatible encoding techniques that strike a balance between efficiency and representational fidelity.

A major limitation shared by both fully quantum and hybrid quantum-classical machine learning (QCML) models [61–65] is their reliance on quantum hardware not only during training but also at inference time. This dependency significantly hinders real-world deployment, as access to quantum computing resources remains limited and costly, even through cloud-based platforms. To overcome this constraint, the Quantum-Train (QT) framework has been introduced [11, 66–76]. QT proposes a shift in how quantum computation is utilized, rather than directly processing data, the quantum model is tasked with generating the weights of a classical neural network. Once training is complete, inference is performed entirely on a classical system using the generated parameters. This removes the need for quantum data encoding and eliminates any quantum resource requirement during inference, making the approach suitable for near-term applications.

Although QT helps reduce training complexity by offloading parameter generation to quantum circuits, the resulting model retains its full parameter count. As a result, inference time remains unchanged compared to conventional models. In contrast, our proposed QRKD approach incorporates knowledge distillation to reduce both training cost and inference-time complexity. By distilling knowledge into a compact student model, QRKD enables faster and more efficient inference, offering a practical advantage in deployment scenarios.

## B Knowledge Distillation in the Quantum Era

KD has been a widely adopted framework for compressing large neural networks into smaller, more efficient models while preserving performance. As QML evolves, KD naturally extends into hybrid and quantum settings. We identify four emerging categories of quantum-era KD, classified by the model types used for the teacher and student, as illustrated in Fig. 3.

Figure 3: **Categories of Knowledge Distillations in quantum era classified by their teacher and student type.**

**Classical-to-Classical (CC) with Quantum Processing.** This category corresponds to our proposed QRKD framework, in which both the teacher and student models are classical, while quantum resources are integrated into the training process to enhance relational distillation. Quantum circuits are used to map geometric relations between embeddings, into a quantum feature space. This helps the student model learn more nuanced relational structures that are possibly difficult to capture through classical processing alone. Once training is complete, the student model remains fully classical, and inference can be carried out without any reliance on quantum hardware. This approach combines the potential computational advantages of quantum processing during training with the practicality and efficiency of classical inference. To the best of our knowledge, QRKD is the first method demonstrated in this setting.

**Classical-to-Quantum (CQ).** In this setting, a classical teacher model transfers knowledge to a quantum student. The typical goal is to compress a large classical network into a QNN with fewer parameters or reduced circuit depth. This strategy aims to produce compact quantum models that are better suited for near-term quantum hardware. However, since inference must still be performed on a quantum device, this setup poses significant challenges for practical deployment at scale. Existing work has primarily focused on small-scale classification tasks. For instance, some studies have used LLMs as teachers to distill knowledge into QNNs for binary classification [29], while others have explored smaller classical models paired with QNNs for three-class problems [77]. In contrast, our QRKD framework addresses a more complex task: sequence-level text generation, which remains largely unexplored in quantum knowledge distillation.

**Quantum-to-Classical (QC).** In this paradigm, a quantum model serves as the teacher, and knowledge is distilled into a classical student model. The objective is to approximate the behavior of a quantum function or decision boundary using a classical network, enabling efficient inference without requiring quantum hardware. This approach is particularly useful when quantum models exhibit strong performance during training but are impractical to deploy due to hardware limitations. Although there is no direct work in the literature explicitly framed as quantum-to-classical knowledge distillation, this direction is conceptually related to the broader field of efficient classical simulation of quantum systems. Several studies have explored this connection through tensor network approximations and neural-network-based quantum state representations [78–80], which share a similar motivation of translating quantum behavior into a classically manageable form.

**Quantum-to-Quantum (QQ).** In the fully quantum setting, both teacher and student are QNNs. Distillation is used to transfer knowledge from a high-capacity or deep quantum model to a more hardware-efficient quantum student, often with fewer qubits or reduced circuit depth [81]. While this strategy is promising for future quantum-native pipelines, current NISQ-era limitations such as noise and decoherence still restrict its practical usage.

Despite recent advances, existing quantum KD research predominantly focuses on binary or three-class classification tasks and requires quantum computation at inference time, which incurs significant cost and limits scalability. Furthermore, the integration of KD into more complex generative tasks, such as text generation, remains largely unexplored in the quantum context. Our QRKD framework addresses these gaps by enabling classical inference while leveraging quantum advantages during training, and by demonstrating effectiveness on sequence modeling tasks beyond classification.

## C Hyperparameter settings in Experiments

This appendix details the hyperparameter configurations used in our experiments and clarifies the implementation specifics of the QRKD method. To illustrate training dynamics, we also provide the learning curves for both MNIST and CIFAR-10 tasks in Figure 4.

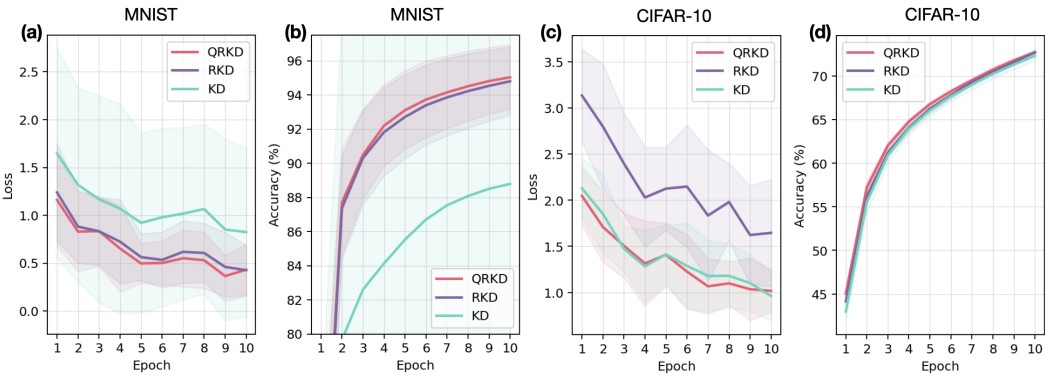

Figure 4: **Training dynamics of QRKD, RKD, and KD methods on MNIST and CIFAR-10 datasets.** (a) and (b) show the training loss and accuracy over 10 epochs on the MNIST classification task, respectively. (c) and (d) illustrate the corresponding trends on the more challenging CIFAR-10 dataset. These results indicate the benefit of incorporating quantum-relational alignment in the knowledge distillation process.

**Software and Hardware.** All experiments were performed on a system equipped with an NVIDIA RTX 3080 Ti GPU. The implementation was based on the TorchQuantum framework [82].

**Optimizer and Learning Rate.** For CNN-based classification tasks (MNIST and CIFAR-10), we used the Adam optimizer with a learning rate of $1 \times 10^{-3}$. For the GPT-2-based text generation tasks (WikiText-2, Penn Treebank, IMDB), we adopted AdamW with a learning rate of $5 \times 10^{-5}$.

**Batch Size and Epochs.** Batch size was set to 64 for classification tasks and 4 for text generation. All CNN models were trained for 10 epochs, while GPT-2 models were fine-tuned for 3 epochs.

**Qubit Count and Repetition.** For all quantum-enhanced methods, the number of qubits used in the feature encoding circuit was fixed at $n_{\text{qubit}} = 4$. The number of repetitions is set to $L_n = 2$ for fidelity kernel, and set to $L_n = 1$ for projected kernel.

**Quantum Kernel Sampling.** To mitigate the computational overhead introduced by quantum operations, the quantum kernel loss $\mathcal{L}_{\text{qr}}$ is estimated on a subset of feature representations within each batch. The sampling strategies are adapted to the structure and dimensionality of the tasks.

For the MNIST classification task, the intermediate feature tensor of shape (64, 12, 4, 4), corresponding to a batch size of 64, is reshaped to (4, 16, 12, 4, 4). We then average along the second dimension to obtain 4 representative samples of shape (12, 4, 4). This setup yields $4 \times 4 = 16$ possible feature pairs $(x_i, x_j)$, from which 4 are randomly selected to compute the quantum kernel values $k(x_i, x_j)$. Each sample is subsequently flattened into a vector of dimension $12 \times 4 \times 4 = 192$. For encoding into a 4-qubit quantum circuit, the input is evenly partitioned such that each qubit processes 48 values, requiring 48 $R_y$ rotation gates per qubit and a total of 192 $R_y$ gates per repetition for the circuit.

For the CIFAR-10 classification task, the procedure mirrors that of MNIST. The intermediate feature tensor of shape (64, 32, 8, 8) is reshaped to (4, 16, 32, 8, 8), and averaged along the second dimension, yielding 4 feature samples of shape (32, 8, 8). This setup yields 16 possible feature pairs $(x_i, x_j)$, and all of them are used to compute the quantum kernel values. Each feature tensor is flattened into a vector of dimension $32 \times 8 \times 8 = 2048$, which is evenly divided across 4 qubits, resulting in 512 $R_y$ rotation gates per qubit and a total of 2048 $R_y$ gates per repetition for the circuit.

For the text generation tasks (WikiText-2, PTB, IMDB) with GPT-2, the final hidden states have shape (4, 512, 768), where 4 is the batch size, 512 is the sequence length, and 768 is the hidden size. These are reshaped into a matrix of shape (49152, 32), and 4 samples are randomly selected for quantum kernel computation. Each selected vector of shape (1, 32) is then encoded into a 4-qubit circuit, requiring 8 $R_y$ rotation gates per qubit. Despite the minimal number of evaluated pairs, the results demonstrate the effectiveness of QRKD even with highly sparse quantum kernel sampling.

Despite the limited sampling, only a small subset of the available pairwise combinations, the empirical results demonstrate that this strategy is effective. This suggests that even sparse quantum relational supervision can provide meaningful regularization and improve student model performance without incurring substantial quantum resource costs.

**Loss Coefficients.** The weighting coefficients $\alpha, \beta, \gamma_d, \gamma_a, \omega$ in the total loss function (see Eq. 14) control the contribution of each component. The specific values used for each method and task are detailed in Table 5 (CNN-based tasks) and Table 6 (GPT-2 tasks).

Table 5: Hyperparameter configurations of for training CNN student model across MNIST and CIFAR-10 datasets.

| Hyperparameters | | F. scratch | KD | RKD | QRKD | QRKD-A | QRKD-D | QRKD-Q |
|---|---|---|---|---|---|---|---|---|
| Optimizer | | | | | Adam | | | |
| LR | | | | | 1e-3 | | | |
| Batch size | | | | | 64 | | | |
| Epochs | | | | | 10 | | | |
| $\alpha$ | (task coeff.) | 1 | 0.5 | 0.5 | 0.5 | 0.5 | 0.5 | 0.5 |
| $\beta$ | (KD coeff.) | 0 | 0.5 | 0.5 | 0.5 | 0.5 | 0.5 | 0.5 |
| $\gamma_d$ | (DR coeff.) | 0 | 0 | 0.1 | 0.1 | 0 | 0.1 | 0 |
| $\gamma_a$ | (AR coeff.) | 0 | 0 | 0.1 | 0.1 | 0.1 | 0 | 0 |
| $\omega$ | (QR coeff.) | 0 | 0 | 0 | 0.1 | 0 | 0 | 0.1 |
| $n_{qubit}$ | | - | - | - | 4 | 4 | 4 | 4 |

# D   Quantum Kernels with Higher Dimension

In the main text, empirical results were obtained using a fixed qubit count of $n_{qubit} = 4$. In this appendix, we extend the analysis to evaluate the impact of increasing the quantum feature space dimensionality by varying the number of qubits. This allows us to examine the expressivity and stability of quantum kernel relations in higher-dimensional Hilbert spaces.

As illustrated in Fig. 5, we explore the performance of QRKD on MNIST with both Fidelity-based Quantum Kernel (FQK) and Projected Quantum Kernel (PQK) under varying qubit counts $n_{qubit} \in \{2, 4, 6, 8, 12\}$. The left panel shows that both FQK and PQK achieve their highest test accuracy when $n_{qubit} = 12$, marked by star symbols. This suggests that increasing the quantum feature space dimensionality can improve student performance—likely due to the enhanced expressivity of the quantum kernel in capturing relational structure.

Table 6: Hyperparameter configurations of for fine-tuning GPT-2 student model across WikiText-2, PTB, and IMDB datasets.

| Hyperparameters | | FT | KD | RKD | QRKD | QRKD-A | QRKD-D | QRKD-Q |
|---|---|---|---|---|---|---|---|---|
| Optimizer | | | | | AdamW | | | |
| LR | | | | | 5e-5 | | | |
| Batch size | | | | | 4 | | | |
| Epochs | | | | | 3 | | | |
| $\alpha$ | (task coeff.) | | | | 1 | | | |
| $\beta$ | (KD coeff.) | 0 | 0.5 | 0.5 | 0.5 | 0.5 | 0.5 | 0.5 |
| $\gamma_d$ | (DR coeff.) | 0 | 0 | 0.1 | 0.1 | 0 | 0.1 | 0 |
| $\gamma_a$ | (AR coeff.) | 0 | 0 | 0.1 | 0.1 | 0.1 | 0 | 0 |
| $\omega$ | (QR coeff.) | 0 | 0 | 0 | 0.1 | 0 | 0 | 0.1 |
| $n_{qubit}$ | | - | - | - | 4 | 4 | 4 | 4 |

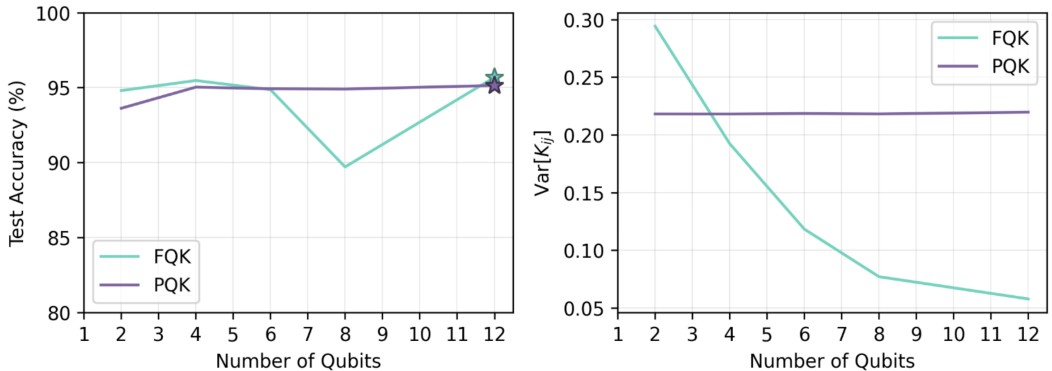

Figure 5: Variance of quantum kernel values with normalized student and teacher feature. [MNIST]

On the right, we analyze the variance of the quantum kernel values. In the FQK case, we observe a rapid decrease in variance as the number of qubits increases, consistent with prior observations in Sec. 2.3 regarding the over-smoothing effect in high-dimensional quantum Hilbert spaces. In contrast, PQK maintains a more stable variance profile across different qubit counts, suggesting that the projection mechanism mitigates the collapse of kernel diversity. This empirical evidence confirms the benefit of projected measurements in preserving meaningful relational information.

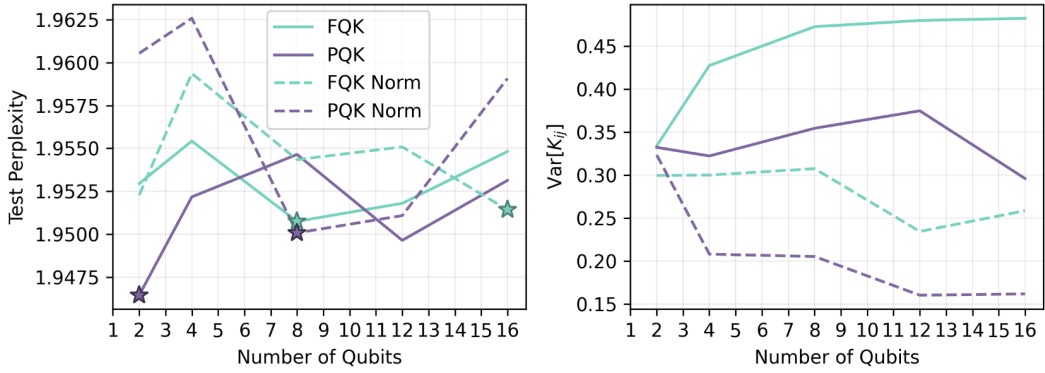

Figure 6: Variance of quantum kernel values with normalized student and teacher feature. [GPT-2]

We further investigate the impact of quantum feature space dimensionality on the GPT-2 text generation task using the WikiText-2 dataset, as shown in Fig. 6. In addition to evaluating FQK and PQK, we consider variants where the input features are normalized before being encoded into the quantum

circuit. Unlike the MNIST case, no monotonic improvement is observed with increasing qubit count. Instead, each setting exhibits an optimal qubit number, suggesting a task-dependent trade-off between expressivity and overfitting in high-dimensional quantum feature spaces.

The right panel reveals distinct trends in kernel variance. When input features are normalized, increasing qubit count leads to a gradual decrease in variance. In contrast, unnormalized inputs exhibit rising variance with more qubits. This phenomenon can be attributed to the magnitude of the input values: unnormalized features tend to have small magnitudes near zero, which results in very small rotation angles in the quantum encoding. These subtle rotations fail to significantly perturb the initial quantum state, causing kernel values to become highly sensitive to slight input differences, thus increasing variance. As the number of qubits grows, the number of rotational gates per qubit decreases, amplifying this effect. Conversely, normalization scales up the inputs, allowing the quantum state to explore a broader region of Hilbert space, which stabilizes kernel outputs and reduces variance.

Overall, these results suggest that (1) increasing qubit count can be beneficial up to a task-specific threshold, and (2) input normalization plays a critical role in controlling kernel variance, particularly in high-dimensional feature regimes. Notably, PQK continues to demonstrate superior stability across configurations, underscoring its practicality for real-world quantum-enhanced distillation.

## E   Is a Quantum Approach Necessary?

The proposed QRKD framework leverages quantum circuits to encode intermediate neural features into high-dimensional Hilbert spaces. The key appeal of using quantum circuits lies in their ability to construct feature maps into exponentially large spaces with a polynomial number of qubits and operations. This property is particularly valuable when the goal is to extract rich similarity structures, such as pairwise relational information, using quantum kernel evaluations.

In principle, quantum kernels offer a computational advantage when their values are hard to approximate classically. For instance, circuits with deep entangling layers or highly non-local operations are believed to generate kernel functions that are intractable to simulate or estimate with classical resources[13, 15], especially when scaling to higher qubit counts, or related to a circuit of time evolution of a chaotic system. This opens up the potential for QRKD to benefit from quantum advantage, particularly in regimes where classical relational measures are insufficiently expressive or computationally expensive.

Nonetheless, it is important to recognize that classical kernel methods have also been explored for knowledge distillation [20]. Several recent works incorporate classical similarity measures, such as Gaussian kernels or neural tangent kernels, into the student training process. A promising direction for future research involves a systematic comparison between classical and quantum kernel-based relational distillation, assessing not only empirical performance but also computational trade-offs and theoretical limits.

Finally, we note that within the current experimental setting, where shallow circuits and few qubits are used, QRKD remains within the classically simulatable regime. In such cases, our method may be interpreted as a quantum-inspired approach. However, as hardware advances and circuits with deeper entanglement become feasible, the same framework may transition into a classically intractable regime, thereby unlocking the full benefits of quantum computation for model compression and distillation.

## F   Relation to Johnson–Lindenstrauss Lemma

The Johnson–Lindenstrauss (JL) lemma [83, 84] asserts that any set of $N$ points in a high-dimensional space can be embedded into a lower-dimensional space of dimension $k = \Omega(\log N/\epsilon^2)$ such that all pairwise distances are approximately preserved with high probability. Specifically, given $0 < \epsilon < 1$ and a set $X$ of $N$ points in $\mathbb{C}^d$, there exists a linear mapping $f : \mathbb{C}^d \to \mathbb{C}^k$ satisfying

$$(1-\epsilon)|u-v|^2 \leq |f(u)-f(v)|^2 \leq (1+\epsilon)|u-v|^2, \tag{15}$$

for all $u, v \in X$. Such a projection can be implemented by generating a random matrix $A \sim \mathcal{N}(0,1)^{k \times d}$, where each entry is independently sampled from a standard normal distribution, and

defining the projection matrix as $P := A/\sqrt{k}$. The JL lemma has become a foundational tool in classical machine learning, with widespread applications in dimensionality reduction, approximate nearest neighbor search, and efficient large-scale data analysis.

This lemma offers an intriguing perspective: given access to quantum state vectors in a high-dimensional Hilbert space $\mathbb{C}^d$, can we apply a JL-like random projection to reduce them to a lower-dimensional space $\mathbb{C}^k$ and compute the kernel in this compressed space, instead of performing computations in the original high-dimensional space? If feasible, this would enable the alignment between student and teacher kernel values to be preserved in the lower-dimensional space $k$.

Assuming the quantum states $|\phi_i\rangle$ derived from both the student and teacher models are projected, we analyze the effect of this projection on the fidelity kernel value:

$$\left| K_{ij} - K_{ij}^{\text{proj}} \right| = \left| |\langle\phi_i|\phi_j\rangle|^2 - |\langle\phi_i^{\text{proj}}|\phi_j^{\text{proj}}\rangle|^2 \right| \le \varepsilon, \tag{16}$$

where $K_{ij}$ denotes the original fidelity kernel value and $K_{ij}^{\text{proj}}$ is the kernel computed after projection. We now derive the deviation between the teacher and student kernel values:

$$\begin{aligned}
&\|K_{ij}^s - K_{ij}^t\| \\
&\le \|K_{ij}^s - K_{ij}^{s,\text{proj}}\| + \|K_{ij}^{s,\text{proj}} - K_{ij}^{t,\text{proj}}\| + \|K_{ij}^{t,\text{proj}} - K_{ij}^t\| \quad \text{(Triangle Inequality)} \\
&\le \|K_{ij}^{s,\text{proj}} - K_{ij}^{t,\text{proj}}\| + 2\varepsilon
\end{aligned}$$

where $K_{ij}^s$ and $K_{ij}^t$ denote the student and teacher kernel values in the original space, and the superscript "proj" indicates their projected counterparts.

This result shows that the kernel alignment error in the original space is bounded by the alignment error in the projected space, plus an additional error term from projection. According to the JL lemma, this projection error $\varepsilon$ decreases as the projected dimension $k$ increases, and vanishes as $k \to \infty$.

In the following sections, we empirically examine how the projection error $\varepsilon$ varies with different projected dimensions, and more importantly, how it impacts the downstream performance of QRKD.

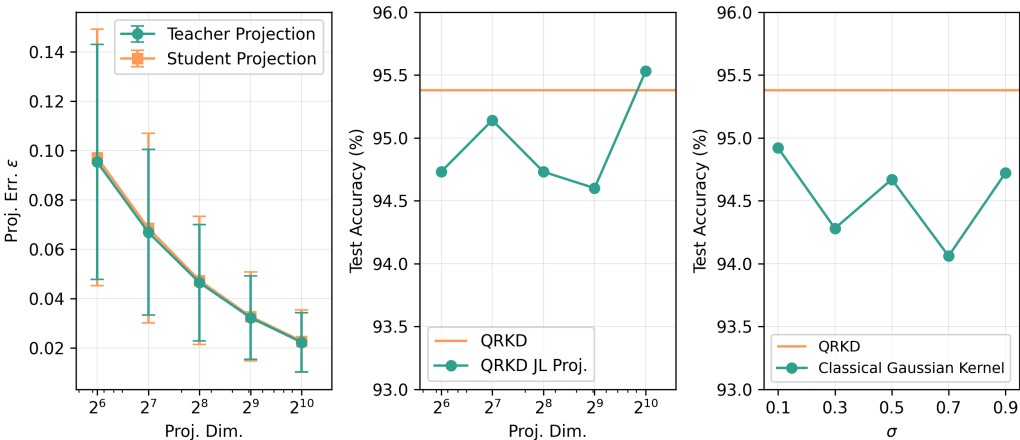

Figure 7: (Left) Projection error $\varepsilon$ for student and teacher quantum state representations under varying projection dimensions. (Middle) Test accuracy of QRKD with JL-projected quantum kernels across projection dimensions. (Right) Comparison between QRKD and classical Gaussian kernel distillation across kernel bandwidth $\sigma$.

**Impact of Projection Error to Lower Dimension.** To understand the effect of projecting high-dimensional quantum states to a lower-dimensional subspace, we examine how the projection error $\varepsilon$ (Eq. 16) behaves under varying projection dimensions. We use the MNIST task with QRKD as an example, using 12 qubits to encode the features, thus the Hilbert space size is $2^{12} = 4096$. After obtaining the quantum states in $2^{12}$ dimensions, we apply the JL-like projection to make the quantum state vector projected to lower dimensions $\in \{2^6, 2^7, 2^8, 2^9, 2^{10}\}$, and calculate the kernel values in the lower dimensions, and use this information for teacher and student alignment in QRKD.

As shown in the left panel of Fig. 7, both teacher and student state representations exhibit rapidly decreasing projection error as the projected dimension increases, consistent with theoretical predictions from the Johnson–Lindenstrauss lemma. Notably, the error remains below 0.02 once the projection dimension reaches $2^{10}$. The middle panel further demonstrates that even with this dimensionality reduction, the test accuracy of the QRKD using JL-projected quantum kernels remains stable and comparable to the original unprojected 12-qubit QRKD baseline, and even surpasses the unprojected baseline when projected to $2^{10}$ dimensions. This indicates that JL-like projections can preserve essential geometric relationships between quantum embeddings, enabling efficient computation of kernel values in lower-dimensional spaces without significant loss of performance.

However, from a practical standpoint, even if low-dimensional projections could make kernel computations more efficient, the main limitation is the need to access the full high-dimensional quantum state. For example, if the quantum state is a highly entangled 100-qubit system, the current formulation would require access to the entire $2^{100}$-dimensional state vector in order to perform the projection. In contrast, the kernel value (such as fidelity) can still be estimated directly through measurements on a 100-qubit quantum computer, provided such a utility-scale device exists.

**Comparison to Classical Kernel Method.** We further compare the performance of our QRKD approach with a classical kernel-based alternative, where all quantum kernels in the distillation process are replaced with classical kernels computed using the radial basis function (RBF), or Gaussian kernel. Here the QRKD result is also the 12-qubit result as in the previous paragraph. The Gaussian kernel between two vectors $x_i$ and $x_j$ is defined as

$$k(x_i, x_j) = \exp\left( -\frac{\|x_i - x_j\|^2}{2\sigma^2} \right), \tag{17}$$

where $\sigma$ is the kernel bandwidth hyperparameter controlling the locality of similarity. As shown in Fig. 7, we sweep across a range of $\sigma \in \{0.1, 0.3, 0.5, 0.7, 0.9\}$, and observe that the performance of the classical Gaussian kernel varies across different values but consistently underperforms the quantum-based QRKD method. This suggests that while classical similarity measures may capture some relational structure, they fail to match the usefulness and representational capacity provided by the quantum feature space, even when the best-tuned Gaussian kernel is used. These findings reinforce the practical value of quantum kernels in relational distillation.

