# OpenReview forum: "Quantum Relational Knowledge Distillation"
_NeurIPS.cc/2025/Workshop/UniReps — UniReps2025_

### Official Review · Reviewer_5o6G · 2025-09-09
**Quantum Relational Knowledge Distillation**

**Confidence:** 5

**Review:**

Quality of the study

The study proposes Quantum Relational Knowledge Distillation (QRKD), which extends classical Relational Knowledge Distillation (RKD) by embedding intermediate teacher and student representations into a quantum Hilbert space using parameterised quantum circuits. The work is technically sound, with clear derivations showing how kernel alignment in Hilbert space ensures functional closeness in Equation 13.
​Experiments are carefully executed across vision (MNIST, CIFAR-10) and language modelling (WikiText-2, Penn Treebank, IMDB with GPT-2). Results consistently show QRKD outperforming KD and RKD in accuracy or perplexity, though the gains are sometimes marginal (especially on CIFAR-10). The ablations (QRKD-A, QRKD-D, QRKD-Q) provide valuable insights into the relative contributions of angular, distance, and quantum relational losses.


Clarity

Firstly, the introduction and motivation are substantial. The gap between KD, RKD, and quantum kernel expressivity is clearly established.
Figures (e.g., Fig. 1) effectively illustrate the QRKD pipeline.

Secondly, the theory section is mathematically correct but dense. Additional diagrams or intuition (e.g., how quantum feature geometry differs from classical RKD geometry) would improve readability for a broader NeurIPS audience.

Lastly, the appendices provide a detailed quantum background, which is helpful for readers unfamiliar with quantum computing.

Originality
To my knowledge, this is the first demonstration of classical-to-classical KD enhanced by quantum relational supervision. Prior quantum KD work has primarily focused on quantum inference ( quantum student models), whereas QRKD cleverly leverages quantum kernels only during training while keeping inference purely classical.

Theoretical contributions extend RKD’s Lipschitz-based analysis to quantum kernel alignment, providing principled justification for improved student generalisation.

Significance

Scientific significance. The work proposes a novel hybrid KD framework that could open a new research direction at the intersection of representation alignment and quantum kernels. It meaningfully extends existing theory (RKD) and shows practical advantages.

Practical significance. Since inference remains classical, QRKD avoids the usual deployment issues of QML, making it more deployable in the near term.

Limitations: The empirical benefits are modest, especially in more challenging vision tasks. Moreover, the reliance on simulated quantum circuits means hardware viability remains unproven.

**Score:**

4

**Topic Fit:**

2

---

### Official Review · Reviewer_cUqC · 2025-09-15
**Novel paper, but this method seems to have diminishing gains on large scale model.**

**Confidence:** 3

**Review:**

This work proposes a novel model distillation method — Quantum Relational Knowledge Distillation (QRKD). QRKD consists of three losses: angle loss, distance loss, and quantum loss. Different from conventional relational knowledge distillation, QRKD enforces functional alignment between teacher and student by matching quantum-encoded structures, which offers greater expressiveness and flexibility. The authors evaluate the method across vision and text modalities: MNIST and CIFAR-10 for CNNs, and WikiText-103, Penn Treebank, and IMDB for GPT-2.

The contribution is original and the approach is novel in introducing quantum encoding to relational distillation. The experimental evaluation is reasonably comprehensive, covering different model sizes and architectures (CNN, ViT) on multiple vision and language datasets.

On smaller datasets such as MNIST, QRKD outperforms baseline methods by a notable margin. However, this improvement does not appear to generalize to larger datasets and models. For CIFAR-10 the gains diminish compared to MNIST, and for GPT-2 QRKD fails to outperform standard KD on perplexity (though QRKD-Q shows a slight improvement). Given these results, I recommend reformulating the overall distillation objective and further studying why performance degrades at scale.

**Score:**

2

**Topic Fit:**

3

---

### Official Review · Reviewer_abwJ · 2025-09-15
**Quantum knowledge distillation in the teacher model improves accuracy in the student model**

**Confidence:** 2

**Review:**

summary:
The paper addresses the problem of knowledge distillation (KD): how do we compress large models when memory is one of the greatest bottlenecks in devices? KD is a method where a large source LLM model (the "teacher") labels data for a smaller model (the "student") to train on. One flaw in this method is that the student model often does not learn the internal representational structures that the teacher model has. RKD is a framework which extracts the relational structure information from the teacher model and transfers it to the student model.
The contribution outlined in the paper, QRKD, is a framework based off of RKD, where the feature space of the teacher model is mapped to quantum states via a parametrized quantum circuit, which allow the teacher model to gain better representational flexibility. The student model is left as a classical architechture, so no quantum machines are needed at inference time. Experiments show that QRKD consistently outperforms KD and RKD methods, even if the difference is slight depending on the task, showing that it is a viable KD method.

These questions remained in my mind as I was reading:
- Why is the difference of internal representational structure between the teacher and student model a bad thing?
- What does "representational flexibility" mean? How does changing the representational structure in the teacher model improve the performance of the student model, especially if the student model is still learning through the classical architecture? I found this point the most fascinating, and would have liked to see more focus on it!
- What do the training times look like? Is it comparable/competitive to classical RKD methods?
- Is the model compression affected by QRKD? Since the original purpose of KD is tackling model storage issues, I found myself expecting discussion on how QRKD performs in that aspect over representational flexibility and accuracy.

I greatly appreciated the thorough and detailed test reports done on various tasks across various KD methods, as well as the discussion on how quantum computing would improve the compromise between representational richness vs model size. In my limited experience it is rather infrequent that ML papers are able to discuss flexibility or range of application in their results, since most ML models and techniques are designed to be specialized to certain tasks. I think clarity and originality are this paper's biggest strengths.

**Score:**

4

**Topic Fit:**

3